# Are Patients with Axial Spondyloarthritis Who Were Breastfed Protected against the Development of Severe Disease?

**DOI:** 10.3390/jcm12051863

**Published:** 2023-02-27

**Authors:** Sara Alonso, Ignacio Braña, Estefanía Pardo, Stefanie Burger, Pablo González del Pozo, Mercedes Alperi, Rubén Queiro

**Affiliations:** 1Rheumatology Service, Hospital Universitario Central de Asturias, 33011 Oviedo, Spain; 2ISPA Translational Immunology Section, Biohealth Research Institute of the Principality of Asturias (ISPA), 33011 Oviedo, Spain; 3School of Medicine, Oviedo University, 33011 Oviedo, Spain

**Keywords:** axial spondyloarthritis, breastfeeding, disease activity, quality of life, prognosis

## Abstract

Background and aims: Breastfeeding is recognized as one of the most influential drivers of the gut microbiome. In turn, alterations in the gut microbiome may play a role in the development and severity of spondyloarthritis (SpA). We aimed to analyze different disease outcomes in patients with axial SpA (axSpA) based on the history of breastfeeding. Patients and methods: A random sample was selected from a large database of axSpA patients. Patients were divided based on history of breastfeeding and several disease outcomes were compared. Both groups were also compared based on disease severity. Adjusted linear and logistic regression statistical methods were used. Results: The study included 105 patients (46 women and 59 men), and the median age was 45 years (IQR: 16–72), and the mean age at diagnosis was 34.3 ± 10.9 years. Sixty-one patients (58.1%) were breastfed, with a median duration of 4 (IQR: 1–24) months. After the fully adjusted model, BASDAI [−1.13 (95%CI: −2.04, −0.23), *p* = 0.015] and ASDAS [−0.38 (95%CI: −0.72, −0.04), *p* = 0.030] scores were significantly lower in breastfed patients. Forty-two percent had severe disease. In the adjusted logistic model for age, sex, disease duration, family history, HLA-B27, biologic therapy, smoking, and obesity, breastfeeding had a protective effect against the development of severe disease (OR 0.22, 95%CI: 0.08–0.57, *p* = 0.003). The selected sample size was sufficient to detect this difference with a statistical power of 87% and a confidence level of 95%. Conclusion: Breastfeeding might exert a protective effect against severe disease in patients with axSpA. These data need further confirmation.

## 1. Introduction

The spondyloarthritis (SpA) concept includes a group of entities with their own clinical-radiological characteristics and a genetic link through HLA-B27. Currently, this nosology family includes entities with inflammatory symptoms and signs that predominate in the axial skeleton, such as radiographic (also known as ankylosing spondylitis -AS-) and non-radiographic axial SpA (axSpA), as well as predominantly peripheral forms such as psoriatic arthritis, among others [1].

The pathogenesis of these entities is only partially known, and although the best-known element of genetic predisposition is HLA-B27, not all patients with SpA express this genetic biomarker [1]. The imbalance between the different species of bacteria, viruses and fungi that colonize the human intestine, a process called gut dysbiosis, seems to be at the origin of these conditions. Emerging evidence suggests that subclinical gut inflammation in patients with SpA, apparently driven by intestinal dysbiosis, is not the consequence of the systemic inflammatory process but rather an important pathophysiological event actively participating in the origin of the disease [2]. This breakdown of intestinal homeostasis is not only a key early step in the development of SpA, but is also associated with the degree of activity and severity of it [3]. Additionally, both the genetic background (HLA-B27) and the level of disease activity are likely to influence the composition of the gut microbiota of patients with SpA [4]. The potential of a hindering effect of NSAIDs on the achievement of treatment objectives in axSpA has also recently been hypothesized through a negative effect of these drugs on the intestinal microbiota of these patients [5].

The establishment of the gut microbiome during early life is a complex process with lasting implications for an individual’s health. Breastfeeding is recognized as one of the most influential drivers of gut microbiome composition during infancy, an effect on the health of individuals that may extend into the advanced stages of life. Differences in gut microbial communities between breast-fed and formula-fed infants have been consistently observed, and are hypothesized to partially mediate the relationships between breastfeeding and the decreased risk for numerous communicable and noncommunicable diseases in early life [6]. Thus, for example, among children with juvenile idiopathic arthritis (JIA), those breastfed for more than 6 months tend to show fewer joint deformities, less disease activity and better physical function [7]. Furthermore, breastfeeding could have potential preventive effects on the development of certain rheumatic diseases during adulthood. Indeed, one study reported that patients with AS had been breastfed less often than healthy controls. In families where children were breastfed, the patients with AS were less often breastfed than their healthy siblings. In addition, breastfeeding reduced the familial prevalence of AS. Therefore, a breastfeeding-induced protective effect on the occurrence of AS has been suggested [8]. Recently, breastfeeding has also been advocated to have a potential modifiable effect in reducing the risk of rheumatoid arthritis (RA) [9].

Despite this information, very little is known about the role of breastfeeding on disease outcomes in certain rheumatic conditions such as axSpA. In this study, we addressed the potential role of breastfeeding on various clinical outcomes in patients with axSpA. We also investigated whether having received breastfeeding could influence the severity of the disease.

## 2. Patients and Methods

This was a retrospective longitudinal study of outpatients with axSpA fulfilling the ASAS criteria [10]. These patients were treated regularly and according to a standard follow-up protocol in a unit specifically dedicated to axSpA of a rheumatology service of a tertiary care hospital. The patients were informed of the purpose of the study and gave their informed consent to carry it out. The ethical standards of good clinical practice contained in the Declaration of Helsinki were followed at all times. Patients seen for the first time in this unit signed an informed consent that authorizes researchers to use the clinical, analytical and radiographic information collected during follow-up so that it can be used for future research purposes. The anonymity of the patients has been safeguarded at all times. The research protocol was reviewed by a clinical research ethics committee (CEIcPA-ref2020/22).

The generic information collected refers to socio-demographic aspects, lifestyles, and previous medical history. Regarding axSpA, the following are collected: age at onset, symptom duration, family history of disease, comorbid factors (psychological dysfunction, osteoporosis, metabolic syndrome), axial and peripheral clinical manifestations, number of painful joints, number of inflamed joints, enthesitis, dactylitis, extra-articular manifestations (uveitis, psoriasis, inflammatory bowel disease), disease activity according to the Bath Ankylosing Spondylitis Disease Activity Index (BASDAI), and the Ankylosing Spondylitis Disease Activity Score (ASDAS), physical function according to the Bath Ankylosing Spondylitis Functional Index (BASFI), structural damage (sacroiliitis, syndesmophytes, vertebral squaring), and the general disease impact according to the Assessment of SpondyloArthritis international Society-Health Index (ASAS HI) questionnaire. We have previously contrasted the validity and feasibility of these measures under routine clinical conditions [11,12]. Current and past medication in relation to SpA (non-steroidal anti-inflammatory drugs (NSAIDs), conventional disease-modifying antirheumatic drugs (DMARDs), biologic and targeted-specific DMARDs), and the pertinent analytical parameters (HLA-B27, rheumatoid factor (RF), antinuclear antibodies (ANA), erythrocyte sedimentation rate (ESR), and C-reactive protein (CRP)) are also collected. These patients are seen every 3 months when they start systemic therapy or when they have poor symptom control, or every 6 months when they progress satisfactorily, reaching treatment goals (ASDAS remission or low activity). The baseline and follow-up protocol were the same for all patients. The work procedures of this SpA unit are regularly audited and currently have the advanced quality seal awarded by the Spanish Society for Healthcare Quality.

For the purposes of this study, patients were asked about their history of breastfeeding (whether or not they received it, and if so for approximately how long). Information on breastfeeding was only cross-checked with the patients’ mothers when the patients were uncertain how long they had been breastfeeding or whether they had received it. In this case, the patients were asked to cross-reference the requested information with their mothers. This part of the study was included in the information sent to the ethics committee for approval. Only patients with reliable information in this regard were included. This implied that if the patients could not provide this information with certainty because they did not remember and it was impossible to cross-reference this information with their mothers, they were not included in the study. Although this implies a certain inclusion bias, this only happened in five cases. The information to carry out this study was collected between 1 February and 30 June 2022. Furthermore, for the purposes of the study, a random sample of patients was chosen from the database of this monographic unit, selecting one out of every three consecutive patients from this database.

## 3. Statistical Analysis

A descriptive statistical study of all the variables was made, using central and dispersion measures for the quantitative variables, as well as absolute and relative frequencies for the qualitative ones. The use of central tendency measures (mean or median) and their dispersion values was made based on the normality (or not) in the distribution of the data. Categorical variables were compared with a Fisher’s exact test and continuous variables with a Mann-Whitney U test, except for age at diagnosis, which followed a normal distribution and was compared with a Student’s *t*-test. To estimate the crude and adjusted effect of breastfeeding on disease outcomes, linear regressions were used, as when the outcome was a quantitative variable, or logistic in the case of categorical ones. To assess whether the age of onset of the disease was influenced by breastfeeding, the Welch two sample *t*-test was used. A Wilcoxon rank sum test with continuity correction was used to assess the crude effect of breastfeeding and its duration on the different disease outcomes (BASDAI, ASDAS, ASAS HI, structural damage). To estimate the potential independent effect of breastfeeding on the outcomes analyzed, a multiple linear regression adjusted for age, sex, disease duration, family history of SpA, HLA-B27, biological treatment, smoking, and obesity was performed. In addition, the crude and adjusted effect of breastfeeding on the severity of the disease was analyzed. For the purposes of this study, severe disease was defined as the combined outcome of BASDAI > 4 and/or ASDAS > 2.1 + BASFI > 4 + ASAS HI > 5, maintained for at least 6 months each year, during follow-up. Regarding the sample size, the number of randomly selected study subjects was sufficient to detect significant differences of two independent proportions (severe vs. non-severe disease), based on the history of breastfeeding, with a statistical power of 87% and a confidence level of 95%. Statistical analyses were performed with R software version 4.0.2.

## 4. Results

The study included 105 randomly selected patients from a larger database of axSpA, 46 women and 59 men, with a median age of 45 (IQR: 16–72) years, and a mean age at diagnosis of 34.3 ± 10.9 years. Sixty-one patients (58.1%) had a history of breastfeeding, with a median duration of 4 (IQR: 1–24) months. Most patients presented radiographic axSpA (n: 63, 60%). Seventy-eight patients (74.3%) were HLA-B27 positive. Fifty-nine percent of the patients were receiving biological therapies (mostly anti-TNFα) at the time of being included in the study. Most patients were reasonably well controlled with a median ASDAS score of 1.90 (IQR: 0–4.70). Table 1 shows the main characteristics of the study population, both overall and divided according to the history of breastfeeding.

Having received breastfeeding did not influence the age of onset of the disease (*p* = 0.69). In the crude estimate, the history of breastfeeding was associated with lower BASDAI [−1.35 (95%CI: −2.30, −0.40), *p* = 0.006], BASFI [−0.85 (95%CI: −1.79, 0.10), *p* = 0.08], ASDAS [−0.47 (95%CI: −0.82, −0.13), *p* = 0.008], and ASAS HI [−2.01 (95%CI: −3.63, −0.38), *p* = 0.02] scores. After the fully adjusted model, BASDAI [−1.13 (95%CI: −2.04, −0.23), *p* = 0.015] and ASDAS [−0.38 (95%CI: −0.72, −0.04), *p* = 0.030] scores remained significantly lower in breastfed patients. After these adjusted models, there was a trend towards lower ASAS HI values [−1.43 (95%CI: −2.98, 0,11), *p* = 0.07], but just bordering statistical significance. However, the duration of breastfeeding did not influence the above outcomes. There was no relationship between structural damage and breastfeeding. In fact, in the multivariate logistic regression model, the only factors associated with greater structural damage were disease duration [OR 1.10, 95%CI: 1.04–1.17, *p* = 0.003] and smoking [OR 4.23, 95%CI: 1.43–13.53, *p* = 0.011].

Of the 44 patients classified as severe disease, 26 had not been breastfed, while the rest had been (crude OR 0.30, 95%CI: 0.13–0.66, *p* = 0.004). In the adjusted logistic model for age, sex, disease duration, family history, HLA-B27, biologic therapy, smoking, and obesity, breastfeeding maintained a protective effect against the development of severe disease (OR 0.22, 95%CI: 0.08–0.57, *p* = 0.003). The other factors with an independent effect on severe disease were found to be male sex (OR 0.30, 95%CI: 0.11–0.79, *p* = 0.018), family history of disease (OR 0.36, 95%CI: 0.12–0.96, *p* < 0.05), and obesity (OR 4.1, 95%CI: 1.01–19.1, *p* < 0.05). There was a nonsignificant trend toward more severe disease among smokers (OR 2.43, *p* = 0.07).

## 5. Discussion

In this study of a randomly selected population of patients with axSpA, it was observed that patients who were breastfed had significantly lower disease activity indices (BASDAI and ASDAS) and less severe disease. However, we did not detect differences in the age of onset of the disease in relation to breastfeeding or an association between the disease outcomes analyzed and the duration of breastfeeding. Nor did we detect a clear relationship between having been breastfed and structural damage, especially in the form of syndesmophytes development. After introducing the adjusted logistic regression models, it was shown that patients who were breastfed reduced the possibility of severe disease by 78%. In addition, the study sample randomly selected for this study had sufficient statistical power (87%) to detect significant differences between the two groups based on the development of severe vs. non-severe disease.

The benefits of breastfeeding are manifold, including the initial shaping of the gut microbiome in humans, with clear implications for the development and maturation of the immune system. Breast milk is a component of the maternal-mucosal immune system that aids in the development and regulation of both the infant’s innate and adaptive immunity. Furthermore, breast milk provides a host of anti-inflammatory, anti-infectious and tolerogenic products [6,13]. These benefits appear to be related to the ability to protect the individual from the development of communicable and non-communicable diseases later in life. Thus, the protective effect of breastfeeding on the future development of cardiovascular diseases, allergies, asthma, or diabetes, among others, has been reported [6,13]. Moreover, it has been hypothesized that individuals who receive breastfeeding could be protected from the future development of AS, while in the case of RA the data are more controversial [8,14]. In this sense, 58% of our series had been breastfed, while the official rates of breastfeeding in Europe are around 70%, which supports this hypothesis [13]. On the other hand, infants with JIA who are breastfed appear to develop less severe disease than formula-fed infants with the same condition [7], so it is possible that the benefits of breastfeeding may not only refer to the possibility of protecting individuals from the development of certain rheumatic diseases, but also that these benefits can be extended to the natural evolution of the disease once it has already started. To the best of our knowledge, there are no studies that have analyzed activity and severity data in relation to a history of breastfeeding in patients with axSpA, and although breastfeeding did not appear to have a protective effect on the development of the disease in our population since both groups developed the disease at very similar ages, it does seem to be associated with a less active and less severe disease. It is also noteworthy that the protective effect against severe disease detected in our study was independent of the duration of lactation. Therefore, what seems important is breastfeeding itself, regardless of its duration. Our findings are in line with a meta-analysis conducted in patients with RA that suggested that breastfeeding was associated with a lower risk of RA, regardless of whether the breastfeeding time was longer or shorter [14].

It is difficult, in any case, to connect an event very early in an individual’s life, such as having been breastfed, with the development and evolution of a disease, such as axSpA, that will occur decades later and that is surely related to many other predisposing factors [15]. In any case, the connections between alterations in the gut microbiome and the development of SpA are strongly supported by many studies of the last decade [2,3,4,5,16], so it is tentative to speculate on the fact that the alterations in the intestinal microbiota involved in the development of SpA are gestated in a very early stage of life, and that, once developed, remain as a perennial dysbiosis imprint that under certain favorable circumstances (HLA-B27, a second hit phenomenon, etc.) lead to the development of clinically evident disease. Supporting this view, it has been shown that breastfed and bottle-fed rhesus macaque infants developed markedly different immune systems, which remained different 6 months after weaning when the animals began to receive identical diets. In particular, breastfed infants developed robust populations of memory T cells as well as T helper 17 (Th17) cells within the memory pool, whereas bottle-fed infants did not [17]. This last data is interesting, since Th17 cells seem to play an essential role in maintaining homeostasis and other barrier functions of the epithelium of the digestive mucosa [2,18]. The alternative hypothesis is that formula milks might contain products that promote an arthritogenic dysbiosis that, again, under favorable circumstances might lead to the development of clinically evident disease decades later. In any case, delving into the dysbiosis-disease connections that are at the base of the pathogenic theory of the gut-joint axis of SpA is beyond the objectives of this discussion.

A notable fact extracted from the adjusted regression models of this study is that smoking was the most determining factor in relation to the adverse outcomes of the disease. Thus, smokers had, in a statistically significant way, an average of 1.56 points more in the BASDAI estimate, 1.59 points more in the BASFI score, 0.54 points more in the ASDAS, and 2.9 points more in the mean estimate of the ASAS HI. For its part, being a smoker increased the possibility of developing syndesmophytes more than fourfold. Moreover, there was a nonsignificant trend toward more severe disease among smokers. These findings are consistent with previous studies in this regard, and highlight the relevance of this modifiable risk factor in disease management strategies [19,20]. In addition, obesity has been linked with worse outcomes in axSpA. In a recent study, this modifiable comorbidity was significantly associated with worse quality of life, greater impairment of functional ability, and a trend toward worse disease activity [21]. Obesity has been associated with a higher reported BASDAI score, and being overweight or obese was associated with a higher degree of spinal stiffness and number of comorbidities compared to under/normal weight respondents [22]. In line with this, our obese patients had a more than fourfold greater likelihood of having severe disease. This data becomes especially important if we take into account that it has been hypothesized that prolonged breastfeeding constitutes a protective mechanism against obesity by affecting long-lasting physiological changes in liver-to-hypothalamus communication and hypothalamic metabolic regulation [23]. In that sense, we found a higher prevalence of obesity among non-breastfed patients. Therefore, a tentative thought is that part of the protective effects of breastfeeding on the development of axSpA and its severity is mediated by a positive effect of breastfeeding (especially prolonged) in the prevention of obesity in later stages of life. Obesity should therefore be considered as a modifiable risk factor for disease activity within axSpA management, and perhaps one of the best ways to combat this predictor of poor outcome is to promote prolonged breastfeeding.

This study has certain weaknesses that we will comment on. First, it is a cross-sectional observation, although the patients were part of a larger cohort subjected to rigorous follow-up. Second, and despite careful questioning about the history of breastfeeding, a false recollection bias cannot be excluded. On the other hand, the sample size is relatively small, although this did not seem to affect the statistical power of the study in relation to the main outcomes, such as the difference in the proportion of patients with severe vs. non-severe disease after being breastfed. In addition, the structural damage was not estimated through a standard measure such as the mSASSS, but rather through the collection of information on variables such as the degree of sacroiliac involvement, vertebral squaring or syndesmophytes formation. The definition of severe disease is not standard, and this can be criticized, although this definition was based on a combined and sustained outcome at the time of the activity, the functional disability, and the global impact of the disease on the subjects. Among the strengths of the study, it is worth noting the detailed and standardized collection of information within a unit specifically dedicated to the study of axSpA patients and subject to periodic audits. We also highlight that the sample of patients was randomly extracted from a larger database, which reinforces the statistical reliability of the study. In addition, factors that are classically associated with worse outcomes in SpA, such as smoking and obesity [19,20,21,22,24], were also associated with worse clinical outcomes in our series, which contributes to reinforce the validity of our results.

## 6. Conclusions

Our study shows that having received breastfeeding, regardless of its duration, is associated with better outcomes in patients with axSpA. Furthermore, breastfeeding seems to exert a certain protective effect against the development of more severe disease. In addition, this protective effect was maintained after the adjustment for several confounders. These data should be confirmed with larger prospective studies.

## Figures and Tables

**Table 1 jcm-12-01863-t001:** Main characteristics of the disease in the overall study population and according to the history of breastfeeding.

	Breastfeeding: No N = 44	Breastfeeding: Yes N = 61	Overall Study Population N = 105
Age, years (mean ± SD)	44.4 (11.7)	42.5 (13.1)	43.3 (12.5)
Male	56.8%	55.7%	56.2%
Educational level:			
Primary	31.8%	31.1%	31.4%
Secondary	36.4%	32.8%	34.3%
University	31.8%	36.1%	34.3%
Radiographic axSpA	75%	49.2%	60.0%
Non-radiographic axSpA	13.6%	32.8%	24.8%
HLA-B27	79.5%	70.5%	74.3%
Age at diagnosis, years (mean ± SD)	34.8 (10.8)	33.9 (11)	34.3 (10.9)
Family history	45.5%	32.8%	38.1%
Smoking	50%	27.9%	37.1%
Obesity	15.9%	11.5%	13.3%
Comorbidities:			
Diabetes	2.3%	1.6%	1.9%
Dyslipidemia	13.6%	18%	16.2%
Hypertension	6.8%	11.5%	9.5%
Depression	29.5%	9.8%	18.1%
Fibromyalgia	4.5%	6.6%	5.7%
MACEs	6.8%	1.6%	3.8%
Uveitis	18.2%	18%	18.1%
IBD	13.6%	6.6%	9.5%
BASDAI (mean ± SD)	4.41 (2.24)	3.06 (2.53)	3.63 (2.49)
ASDAS (mean ± SD)	2.31 (0.8)	1.84 (0.9)	2.04 (0.90)
ASAS HI (mean ± SD)	6.3 (4.2)	4.3 (4.1)	5.15 (4.2)
BASFI (mean ± SD)	3.2 (2.2)	2.4 (2.5)	2.73 (2.4)
Syndesmophytes (% of patients)	27.3%	19.7%	22.9%
NSAIDs	81.8%	78.7%	80.0%
Biologics	56.8%	60.7%	59.0%

SD: standard deviation; axSpA: axial spondyloarthritis; HLA: human leukocyte antigen; MACEs: major adverse cardiovascular events; IBD: inflammatory bowel disease; BASDAI: Bath ankylosing spondylitis disease activity index; ASDAS: ankylosing spondylitis disease activity score; ASAS HI: Assessment of SpondyloArthritis international Society-Health Index; BASFI: Bath ankylosing spondylitis functional index; NSAIDs: non-steroidal anti-inflammatory drugs. Note: forms with peripheral onset that evolved to axial forms were included in the study, but are not represented in the tables.

## Data Availability

The data on which this study is based are stored in databases. This information is available to third parties on reasonable demand.

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
