# Peer review of "Are Patients with Axial Spondyloarthritis Who Were Breastfed Protected against the Development of Severe Disease?"

_jcm, 2023, doi:10.3390/jcm12051863_

Round 1

Reviewer 1 Report

1.      The introduction correctly ends with the study aims (“the potential role of breastfeeding on various clinical outcomes in patients with axSpA”), but it also concludes the study (“HIGHLIGHTING a possible protective effect of breastfeeding against severe disease”) because of the authors’ choice of verbs. Conclusive remarks are suitable for the conclusion section, not for the introduction/aims section. Therefore, the second part of the cited phrase should be rephrased as the authors see fit in order to remain in the area of hypothesis and objectives.

2.      The first phrase of the “Patients and methods” section has no verb: “Retrospective longitudinal study of patients with axSpA fulfilling the ASAS criteria [10]”. Concision is appreciated, but elliptical, class notes style phrases are not suitable for scientific literature. Please add at least one verb.

3.      Regarding inclusion criteria, we read that patients were recruited from a tertiary rheumatology clinic. What kind of patients were they? In-patients? Out-patients? In-patients tend to have a more severe disease, and severity is analyzed in this study, so this information is important.

4.      Uveitis, psoriasis and inflammatory bowel disease are not “co-manifestations” of axSpA, they are extra-articular manifestations of the disease.

5.      There are several abbreviations (for example DMARDs, NSAIDs, RF, ANA, ESR, CRP) which are not explained in the text. Please update abbreviation management.

6.      Since breastfeeding “information was collated with the patient's mother”, the study actually included both axSpA patients and their mothers. Did the mothers agree to participate? Were they interrogated by the authors or by their children? The mother issue should be briefly addressed in the text.

7.      Breastfeeding “information was collated with the patient's mother” “in most cases”. What does “most” mean? Please update the text with a precise fraction.

8.      The authors state that “clear and reliable” breastfeeding information were used to decide which patients are included and which patients are excluded from the study. What does “clear and reliable” mean? Firstly, did the authors use more specific criteria to take this decision of clarity? Secondly, did the authors use the same questions for all participants? These points should be briefly addressed in the text.

9.      Methodologically correct and very impressive stats, especially since they are done with R. However, R software is not referenced/credited correctly (the authors only mention “R”).

10.  All the references are numbered two times: “3. 3. Sagard J, Olofsson …”.

Author Response

    The introduction correctly ends with the study aims (“the potential role of breastfeeding on various clinical outcomes in patients with axSpA”), but it also concludes the study (“HIGHLIGHTING a possible protective effect of breastfeeding against severe disease”) because of the authors’ choice of verbs. Conclusive remarks are suitable for the conclusion section, not for the introduction/aims section. Therefore, the second part of the cited phrase should be rephrased as the authors see fit in order to remain in the area of hypothesis and objectives.

Response: Thank you very much for this suggestion. We have rephrased the part the reviewer mentions.

    The first phrase of the “Patients and methods” section has no verb: “Retrospective longitudinal study of patients with axSpA fulfilling the ASAS criteria [10]”. Concision is appreciated, but elliptical, class notes style phrases are not suitable for scientific literature. Please add at least one verb.

Response: Thank you again for this suggestion. Correction has been made accordingly.

    Regarding inclusion criteria, we read that patients were recruited from a tertiary rheumatology clinic. What kind of patients were they? In-patients? Out-patients? In-patients tend to have a more severe disease, and severity is analyzed in this study, so this information is important.

Response: Good point. All were out-patients. This is clarified in the new version of the manuscript.

    Uveitis, psoriasis and inflammatory bowel disease are not “co-manifestations” of axSpA, they are extra-articular manifestations of the disease.

Response: Corrected.

    There are several abbreviations (for example DMARDs, NSAIDs, RF, ANA, ESR, CRP) which are not explained in the text. Please update abbreviation management.

Response: All abbreviation are now explained.

    Since breastfeeding “information was collated with the patient's mother”, the study actually included both axSpA patients and their mothers. Did the mothers agree to participate? Were they interrogated by the authors or by their children? The mother issue should be briefly addressed in the text.

Response: We have mis phrased this part of the methodology. The information on breastfeeding was only collated with the patients´ mother when patients were not sure of the time of lactation or of having received it. In that case, the patients were asked to cross-reference the requested information with their mothers. This part of the study was included in the information sent to the ethics committee for approval.

    Breastfeeding “information was collated with the patient's mother” “in most cases”. What does “most” mean? Please update the text with a precise fraction.

Response: Updated.

    The authors state that “clear and reliable” breastfeeding information were used to decide which patients are included and which patients are excluded from the study. What does “clear and reliable” mean? Firstly, did the authors use more specific criteria to take this decision of clarity? Secondly, did the authors use the same questions for all participants? These points should be briefly addressed in the text.

Response: Very important point. This entire part of the methodology has been rephrased for clarity.

    Methodologically correct and very impressive stats, especially since they are done with R. However, R software is not referenced/credited correctly (the authors only mention “R”).

Response: Corrected.

    All the references are numbered two times: “3. 3. Sagard J, Olofsson …”.

Response: Corrected.

Reviewer 2 Report

The authors evaluated the influence of breastfeeding on the severity of ax SpA. They conducted a retrospective longitudinal study involving 105 patients with radiographic and non radiographic ax SpA. Potential factors that can influence disease activity /severity were taken account into multivariable analysis. The severity of the disease was defined by a composite of validated disease activity measurements: ASDAS, BASDAI and also ASAS HI.

The results showed that patients with a past history of breastfeeding had less severe disease compared to those without breastfeeding. The paper is clear and well written. Despite a low number of patients included, the results were significant. I have only a comment: Table 1 describes the patients with radiographic and non radiographic ax SpA. Patients with a "mixed" form (6.7% of the series) are also indicated. What is a "mixed" form ? Patients can be categorized as having a radiographic or non radiographic form, but not a "mixed" or intermediate form. Please clarify

Author Response

The authors evaluated the influence of breastfeeding on the severity of ax SpA. They conducted a retrospective longitudinal study involving 105 patients with radiographic and non radiographic ax SpA. Potential factors that can influence disease activity /severity were taken account into multivariable analysis. The severity of the disease was defined by a composite of validated disease activity measurements: ASDAS, BASDAI and also ASAS HI.

The results showed that patients with a past history of breastfeeding had less severe disease compared to those without breastfeeding. The paper is clear and well written. Despite a low number of patients included, the results were significant. I have only a comment: Table 1 describes the patients with radiographic and non radiographic ax SpA. Patients with a "mixed" form (6.7% of the series) are also indicated. What is a "mixed" form ? Patients can be categorized as having a radiographic or non radiographic form, but not a "mixed" or intermediate form. Please clarify.

Response: We agree. To avoid confusing readers, we have omitted this part of "mixed forms" because in fact the literature does not include this figure. We add a brief legend saying that the remaining percentage refers to forms with a peripheral onset that evolved into axial forms. Thanks for the suggestion.